# Discovery of New Coumarin-Based Lead with Potential Anticancer, CDK4 Inhibition and Selective Radiotheranostic Effect: Synthesis, 2D & 3D *QSAR*, Molecular Dynamics, In Vitro Cytotoxicity, Radioiodination, and Biodistribution Studies

**DOI:** 10.3390/molecules26082273

**Published:** 2021-04-14

**Authors:** Mona O. Sarhan, Somaia S. Abd El-Karim, Manal M. Anwar, Raghda H. Gouda, Wafaa A. Zaghary, Mohammed A. Khedr

**Affiliations:** 1Labelled Compounds Department, Hot Lab Centre, Atomic Energy Authority, Cairo 13759, Egypt; monasarhan@windowslive.com; 2Department of Therapeutic Chemistry, National Research Centre, Dokki, Cairo 12622, Egypt; somaia_elkarim@hotmail.com (S.S.A.E.-K.); manal.hasan52@live.com (M.M.A.); 3Department of Pharmaceutical Chemistry, Faculty of Pharmacy, Helwan University, P.O. Box 11795, EinHelwan, Cairo 13759, Egypt; raghdahassan@yahoo.com

**Keywords:** coumarin, synthesis, molecular dynamics, radioiodination, CDK4

## Abstract

Novel 6-bromo-coumarin-ethylidene-hydrazonyl-thiazolyl and 6-bromo-coumarin-thiazolyl-based derivatives were synthesized. A quantitative structure activity relationship (*QSAR)* model with high predictive power r^2^ = 0.92, and RMSE = 0.44 predicted five compounds; **2b**, **3b**, **5a**, **9a** and **9i** to have potential anticancer activities. Compound **2b** achieved the best ΔG of –15.34 kcal/mol with an affinity of 40.05 pki. In a molecular dynamic study **2b** showed an equilibrium at 0.8 Å after 3.5 ns, while flavopiridol did so at 0.5 Å after the same time (3.5 ns). **2b** showed an IC_50_ of 0.0136 µM, 0.015 µM, and 0.054 µM against MCF-7, A-549, and CHO-K1 cell lines, respectively. The CDK4 enzyme assay revealed the significant CDK4 inhibitory activity of compound **2b** with IC_50_ of 0.036 µM. The selectivity of the newly discovered lead compound **2b** toward localization in tumor cells was confirmed by a radioiodination biological assay that was done via electrophilic substitution reaction utilizing the oxidative effect of chloramine-t. **^131^I-2b** showed good in vitro stability up to 4 h. In solid tumor bearing mice, the values of tumor uptake reached a height of 5.97 ± 0.82%ID/g at 60 min p.i. **^131^I-2b** can be considered as a selective radiotheranostic agent for solid tumors with promising anticancer activity.

## 1. Introduction

Cancer disease is the second largest cause of death globally. Based on the WHO survey, around 9.6 million deaths occurred in 2018 and is predicted to reach 13.1 million by 2030 specially in children [1,2,3,4,5]. Inhibition of cyclin-dependent kinases (CDKs) is a promising target for treating cancer [6,7,8,9,10,11,12,13,14,15]. CDK4 is a key factor in initiation and promoting of various tumors due to its overexpression in different tumor cells [16,17,18]. CDK4 inhibition is featured as a potential strategy for targeted treatment of several cancers leading to cell cycle arrest within the G1 phase 1 [19,20]. During the last decade, three generations of CDK inhibitors, have been developed [21] (Figure 1). The absence of appropriate balance between the efficiency and the safety of the first-generation pan-CDK inhibitors, such as flavopiridol [22], and seliciclib [23], because of their multi-target activity was the main reason for the development of more selective CDK inhibitors, such as milciclib [24], dinaciclib [25], CYC-065 [26], and AT7519 [27] (Figure 1). In comparison with non-specific CDK suppressors, CDK4/6 inhibitors prevent the off-target toxicity and produce a specified therapeutic window since they do not inhibit the CDKs that regulate the normal cell cycle [28]. Accordingly, the recent research has been directed towards the third-generation CDK4/6 inhibiting drugs, palbociclib, ribociclib, and abemaciclib [29,30,31,32]. However, these drugs lack of sensitivity to CDK4/6 monotherapy [33,34].

Chromene and its isosteric scaffold coumarin are considered to be the most versatile chemical rings for design and discovery of potential anticancer drugs. Flavopiridol (alvocidib; Figure 1 and Figure 2) is a semisynthetic flavone showing modest potency and selectivity for CDKs 1, 2, 4, 6, 7, and 9. Treatment of cells with flavopiridol leads to delay in S-phase, followed by arresting the cell cycle at G2 phase [35]. Flavopiridol **I** had been utilized in multiple clinical trials, as a monotherapy and in combinations, but with detected low activity and high toxicity [35,36]. In vitro biological studies showed that the chromene–triazole–coumarin triads **II** and **III** was potential fluorescent suppressors of CDK2/CDK4 induced tumors with potent activity against human cervical cancer cell line (HeLa) (Figure 2). Both molecules represented high selectivity towards CDK2 and CDK4. The detected significant binding energies of the triads were referred to their strong hydrophobic interaction between the target proteins and the bulky coumarin-triazole-chromene moieties. This supported them to be encapsulated into the hydrophobic pockets of CDK2 and CDK4. Furthermore, there is also a possibility of hydrogen bonding and π–cation interactions between the inhibitor molecules and CDK4 enhancing the binding affinities [37]. A recent study showed that the newly synthesized 4-flourophenylacetamide-acetyl coumarin (4-FPAC) **IV** produced cytotoxic effect and arrested metastasis in A549 as well as cell cycle progression at the G0/G1 by modulating p21, CDK2, and CDK4 expression [38] (Figure 2). 

Furthermore, the thiazole moiety is used in the discovery of new lead antitumor compounds [39]. Banyu Tsukuba Research Institute, Banyu Pharmaceutical Co., Ltd (Tsukuba, Japan) applied various structural modifications [40] in order to discover the 2-aminothiazole compounds **Va**, **Vb**, and **Vc** as selective and potent inhibitors of CDK4 with IC_50_ values of 4.2, 34, and 9.2 nmol/L, respectively exhibiting potent antiproliferative activities against various cancer cell lines (Figure 3). The optimization studies showed that the compounds **VIa**, **VIb**, **VIc** exhibited potent antiproliferative activity against HCT-116 and PC-6 cancer cell lines with potent selectivity [41]. Moreover, Tadesse and his team work developed the thiazole product **VIIa** which had a potent activity on CDK4 and CDK6 (*K*_i_ = 7 and 42 nmol/L, respectively). Additionally, **VIIa** exhibited antiproliferative activity against a panel of human cell lines, and arrested the cell cycle at G1 phase in M249 and M249R melanoma cell lines [42]. Further structural optimization led to the formation of the thiazole-pyrimidine compounds **VIIb** and **VIIc** which remained as CDK4/6 inhibitors (*K*_i_ = 4/30 and 2/55 nmol/L, respectively) with selectivity over CDKs 1, 2, 7, and 9 [43] (Figure 3). Other modifications showed that the two *N*-aryl aminothiazoles **VIII**, **IX** produced in vitro CDK4 inhibition activity with IC_50_ = 26 nM and IC_50_ = 233 nM, respectively, with in vivo anticancer activity [44] (Figure 3).

The molecular hybridization of two or more bioactive pharmacophores is considered to be an optimistic strategy for the design and discovery of novel anticancer drugs [45,46]. This study has been focused on the development of two new sets of hybridized compounds to be evaluated as potential anticancer agents against some cancer cell lines with possible CDK4 kinase inhibitory activity. The first set (**X**) was bearing 6-bromocoumarin nucleus in conjugation with different substituted thiazolidine/thiazoline/thiazole rings via ethylhydrazone linker which has been confirmed to enhance the antiproliferative activity of various compounds [47,48]. While in the other set (**XI**) the thiazole ring was directly conjugated with 6-bromocoumarin ring at its C-4 position and substituted with various heterocyclic or aromatic rings via the hydrazone linker at its C_2_ position (Figure 4). All the prepared conjugates were subjected to a *QSAR*-based prediction, then examined for their cytotoxic effect against a panel of cancer cell lines by MTT assay. Thereafter, the compounds which showed high prediction with the most efficient activity were investigated for their CDK4 inhibitory activity. Molecular docking and dynamic simulations were applied to evaluate the binding modes and stability in the CDK4 active site in order to justify their inhibitory. The selectivity toward tumor cells can be evaluated by radio-iodination assay [49]. The selectivity of compound **2b** with promising cytotoxic and CDK4 inhibition activity was confirmed by a radio-iodination assay. The radiolabeling study confirmed that **2b** can be used as a potential imaging probe for tumors.

## 2. Results and Discussion

### 2.1. Chemistry

The Synthetic processes of the intermediates and target compounds were accomplished according to the steps illustrated in Scheme 1 and Scheme 2. Characterization and the purity determination of the new target compounds were performed by melting points determination, spectroscopic methods (IR, ^1^H NMR, ^13^C NMR, and mass spectra) in addition to elemental analysis. Compound 3-acetyl-6-bromo-2*H*-chromen-2-one (**1**) was obtained by a Knoevenagel condensation reaction when 6-bromo salicylaldehyde was allowed to react with ethyl acetoacetate in the presence of few drops of piperidine according to the reported method [50,51,52,53]. Furthermore, the intermediates 2-(1-(6-bromo-2-oxo-2*H*-chromen-3-yl)ethylidene)-*N*-methyl (phenyl) hydrazine-1-carbothioamide **2a,b** were prepared by the reaction of **1** with *N*-methyl (phenyl)hydrazine-1-carbothioamide in absolute ethyl alcohol acidified with a catalytic amount of acetic acid in accordance to the documented method [54,55,56]. Thiazolidin-4-one derivatives **3a,b** were obtained via the treatment of the intermediate carbothioamides **2a,b** with ethyl bromoacetate in ethanol containing anhydrous sodium acetate (Scheme 1). IR spectra of the new compounds **3a,b** showed the characteristic absorption bands of 2 C=O of coumarin and thiazolidine rings at 1744, 1700 cm^−1^. While ^1^H NMR spectra of **3a,b** revealed a singlet signal at δ 2.09–2.35 ppm with three protons integration attributed to the hydrazone linker—CH_3_ (another singlet signal at δ 3.20 ppm due to the N-CH_3_ protons of **3a**), a singlet at δ 3.97–4.12 ppm assigned for methylene protons of thiazolidine—CH_2_ as well as different singlet and doublet signals at the range δ 7.42–8.17 ppm related to the aromatic protons. ^13^C NMR spectra of **3a,b** represented two signals at δ 17.30 and 32.6 ppm related to the hydrazone linker–CH_3_ and the thiazolidine-CH_2_ carbons (for **3a**; a third singlet should be appeared δ 29.91 ppm due to N-CH_3_ of thiazolidine ring), the aromatic carbons appeared at δ 116.8–160.9 ppm and the two C=O groups of the coumarin and thiazolidine rings appeared at δ 165.20 and 172.80 ppm. Furthermore, the reaction of the carbothioamides **2a,b** with phenacyl bromide and *p*-bromophenacyl bromide in refluxing ethanol led to the preparation of the corresponding thiazoline compounds **4a,b** and **5a,b** (Scheme 1). IR spectra of the new derivatives **4a,b** and **5a,b** revealed a characteristic absorption band at the range 1744–1728 cm^−1^ assigned for the coumarin-C=O group. ^1^H NMR spectra of **4a,b** and **5a,b** compounds were characterized by the appearance of two singlets at the regions δ 2.07–2.34 ppm and δ 6.46–6.76 ppm displayed for the hydrazone linker—CH_3_ and the methine proton of the thiazoline-H_5_, an additional singlet with three protons integration was recognized at δ 3.31, 3.45 ppm corresponding to (-N-CH_3_) of the compounds **4a**, **5a**, respectively. Also, ^13^C NMR spectral data of compounds **4a, 5a** showed the hydrazone—CH_3_ and -*N*-CH_3_ at δ 16.90, 19.20, 34.11, 35.01 ppm, respectively, the aromatic carbons appeared at δ 100.80–160.0 ppm as well as the coumarin—C=O groups appeared at δ 175.61 ppm.

On the other hand, the target thiazolidinone compounds **6a,b** were obtained via thia-Michael addition reaction between maleic anhydride employed as a Michael acceptor and the semicarbazides **2a,b**, in refluxing toluene (Scheme 1). The IR spectra of both derivatives represented absorption bands at the range 3441–3449 cm^−1^ related to OH groups and at the range 1721–1675 cm^−1^ contributed to C=O groups. Furthermore, ^1^H NMR spectra of compounds **6a,b** revealed the two methylene protons of the acetic side chain as two multiplets at δ 2.87–2.95 and 3.04–3.09 ppm, whereas the triplet signal recognized at 4.39 ppm was conferred to the thiazolidinone-H_5_ proton. The three protons of -CH_3_ of **6a,b** resonated at δ 2.35, 2.09 ppm, respectively, -N-CH_3_ of **6a** at δ 3.20 ppm, the aromatic protons resonated at δ 7.40–8.17 ppm, whereas, the hydroxyl proton appeared as a D2O exchangeable downfield signal at δ 12.75 ppm. ^13^C NMR spectra of **6a,b** represented signals at δ 17.3, 30.0, 37.2, 43.20 ppm due to the carbons of hydrazone-CH_3_, -N-CH_3_ (for **6a**), CH_2_, and thiazolidinone-C5, respectively. The carbonyl carbons appeared as two singlets at the range δ 172.20–174.50 ppm.

The reaction of different aromatic/heterocyclic aldehydes with thiosemicarbazide accomplished the corresponding substituted arylidene-hydrazine-1-carbothioamides **7a**–**I**
^57−63^, which in turn were allowed to react with 6-bromo-3-(2-bromoacetyl)-*2H*-chromen-2-one (**8**) in absolute ethanol to give the corresponding coumarin—thiazole derivatives **9a**–**I** (Scheme 2). 

IR spectra of the compounds **9a**–**i** showed absorption stretching bands within the range of (33418–3503 cm^−1^) and (1675–1728 cm^−1^) contributing to NH and C=O groups, respectively. Furthermore, ^1^H NMR chemical shifts investigated the aromatic protons, the coumarin-H4 and the azomethine proton at the range δ 7.00–8.50 ppm, while the -NH protons resonated downfield as a singlet signals at the range δ 11.97–12.77 ppm (D_2_O exchangeable). In addition, a singlet signal with three protons integration assigned for CH_3_ of compounds **9b**, **9h** at δ 2.33, 2.37 ppm, respectively, whereas the six protons related to -N(CH_3_)_2_ group of the compound **9e** appeared as a singlet signal at δ 3.00 ppm. On the other hand, the six methoxy protons of compound **9c** exemplified two singlet signals at δ 3.80, 3.82 ppm. Also, ^13^C NMR spectra of the compounds **9a**–**i** exhibited the aromatic and -C=O carbons at the region of δ 106.30–175.50 ppm. Furthermore, -CH_3_, -N(CH_3_)_2_, 2OCH_3_ carbons of compounds **9b**, **9h**, **9e**, **9c** were presented as singlets at δ 21.60, 14.21, 41.0, 55.81 ppm, respectively. Mass spectra represented the molecular ion peaks of the new compounds which concurred with their molecular structures.

### 2.2. Quantitative Structure Activity Relationship model (QSAR-Model)

Quantitative structure activity relationship (QSAR) studies are applied to discover the hidden features within the compounds that may be correlated with biological activity [57]. Thiazolyl-hydrazono-coumarin hybrids were reported to have potential anticancer and potent cyclin dependent kinase inhibition activity [47]. Here, the QSAR model was done using this series to be used for prediction of the biological activity of the newly synthesized compounds. The QSAR model was created where the reported 16 thiazolyl-hydrazono-coumarin compounds were used as a training set and our new synthesized compounds (Table 1) were used as a test set. All molecular descriptors available in MOE 2016.08 58 [58] were computed looking for the descriptor/s that will result in the lowest root mean square error (RMSE) and highest coefficient of determination for training set prediction r^2^ that can confirm its powerful predictive power.

In order to discover the descriptor/s that may be correlated with the biological activity, we had to convert all *IC*_50_ biological values of the reported compounds into *pIC*_50_ values to have numerical data that can be handled statistically. Then a database was created with all chemical structures of the previously reported active compounds and their *pIC*_50_ values were added to the database as well. Molecular descriptors in MOE are classified into physical, 2D, and 3D descriptors sets. Each set has a number of related descriptors, as a result each set was evaluated separately. After calculation of each descriptor set for the built database, a correlation plot was created for each descriptor to find out which one will have a direct linear correlation with *pIC_50._* The best linear model should achieve a higher coefficient r^2^ value close to 1.00 and lowest root mean square error (RMSE) value. In our case, the linear model (Figure 5) achieved the best r^2^ = 0.92 and lowest RMSE = 0.44 when BCUT_PEOE (2D descriptors) were applied. The BCUT_PEOE_0 member of these descriptors was irreversibly proportional to the biological activity. This kind of descriptors is responsible for the atomic partial charge distribution. It can have a direct effect on the ability of the compounds to form H-Bonds, and an irreversible effect with the hydrophobic feature of the compounds (highly hydrophobic compounds are less active).

### 2.3. QSAR-Based Prediction of the Anticancer Activity of the New Compounds

Now the QSAR-model is ready to be used for prediction of the anticancer activity of the synthesized compounds by comparing their chemical structures to those in the QSAR-model database then the predictive *pIC*_50_ values were computed to the new compounds and analyzed to select those close to the rage of the active reported compounds so, they were predicted to be active (Table 1). All compounds showed promising predicted activity however, few compounds (**2b**, **3b**, **5a**, **9a**, **9i**) showed predicted *pIC*_50_ values in the range of (8–9) which was the range of the top ranked potent compounds in the previously reported work. As a result, these five compounds were selected for further docking, cyclin-dependent enzyme assay. 

### 2.4. Molecular Docking Results

In a previously reported work [47] thiazol-hydrazono-coumarin hybrids showed promising cytotoxic effect with CDK2 inhibitory activity. In this work we tried to investigate the effect of CDK4 inhibition on the activity of coumarin-based derivatives. In order to perform the docking study we attempted to identify the active site of CDK4 that can be used for ligand docking. Unfortunately, the only reported CDK4 crystal structure in protein data bank has no complexed ligand (pdb code = 2W9Z). Flavopiridol has been reported to inhibit CDK2 and was co-crystallized with CDK2 (pdb code = 6GUB). Sequence alignment between CDK2 and CDk4 in order to determine the identity percentage and it was found to be 42.7%, that can confirm the high identity specially in the binding site residues (Figure 6A) The crystal structure of CDK2 in complex with flavopiridol and CDk4 has been superimposed to identify the binding site of flavopiridol to be used for docking (Figure 6B). Flavopiridol was kept in the binding site of CDK4 and this model was be used for docking. The analysis of the docking (Table 2) showed that compound 2b revealed a hydrogen bond formed between –NH of the side chain and -C=O of Asp99. The presence of the electronegative “S” atom supported the intramolecular hydrogen bonding formation within compound **2b** and helped in the formation of a stable conformation that was able to interact with both Asp99 and Glu144 (Figure 7A). Both C=O group and the oxygen atom of coumarin ring interacted with Val96. In addition, the electron acceptor group -C=N showed interaction with Ala33 (Figure 7B). Hydrophobic interactions formed between the coumarin ring that came in front of the His95 imidazole ring (Figure 7B). Compound **3b** showed hydrophobic interactions with both Ala33 and His95. It also, had a hydrogen bond with Asp158 (Figure 7C). Compound **5a** showed two hydrogen bonds with Asp97 and Glu144 (Figure 7D).

Compound **9a** and **9i** shared a common pose of interactions where they formed two hydrogen bonds with Lys35, a hydrogen bond with Asp158, Ala33, and Val96 (Figure 8A,B). Docking of flavopiridol showed more computed affinity (45.50) and better free energy of binding (−19.91 kcal/mol) than that of compound **2b**. Flavopiridol showed hydrophobic interactions with Ala33, Phe93, Val20, and Ile12 in addition to a hydrogen bond with Asp158 (Figure 8C). In another pose, flavopiridol illustrated a hydrogen bond with -NH of Asp158 and another hydrogen bond that was formed between the oxygen atom of 3-hydroxy group of piperidinyl moiety and the -NH group of His95 (Figure 8D).

### 2.5. Molecular Dynamic Simulations

Molecular dynamic simulation study is usually applied to evaluate the strength and stability of binding of the promising compounds. The top-ranked compound in both; *QSAR*-based prediction, and docking score was **2b**. It was subjected to drastic conditions in which high temperature and pressure in addition to movement and rotation of all bonds and angles were applied over 10 ns period of time. The results of these MD study were compared to the reference flavopiridol (Figure 9). The oscillations of compound **2b** was parallel to that flavopiridol. However, flavopiridol was more stable as its equilibrium was reached at RMSD of 0.5 Å after 3.5 ns. While compound **2b** reached the equilibrium at RMSD of 0.8 Å after the same time 3.5 ns and this was interesting and confirmed the binding stability of our compound. Additionally, it confirmed the previous prediction of the high biological activity of this compound.

### 2.6. Biological Evaluation of the Synthesized Compounds

#### 2.6.1. In Vitro Anticancer Effect and Structure Activity Relationship

The cytotoxic activity of the compounds was tested against three cell lines MCF-7 Cell line (Breast cancer), A-549 (human alveolar basal epithelial cells), and CHO-K1(Ovarian cell line). All the compounds exhibited moderate cytotoxic activity, however, compound **2b** showed the best cytotoxic effect on the tested cell lines (Table 3). Despite the apparent promising activity of the compounds they showed lower activity when compared to flavopiridol.

Studying the effect of structure variation on the anticancer activity (Figure 10), it appeared that the presence of a thiourea moiety had a significant effect on the activity whereas its inclusion in a ring system such as thiazolidinone (**3a**), thiazoline (**4b**) markedly decreased the cytotoxic activity. Substitution on the thiourea group also appeared to possess a major impact on the anti-cancer activity. *N*-phenyl thiourea moiety appeared to have the optimum activity. While the 3-phenylthiazolidin-4-one (**3b**), diphenylthiazoline (**4b**), 3-methyl-4-phenylthiazol (**4a**), 4-bromophenyl-3-phenylthiazol (**5b**), 4-oxo-3-phenylthiazolidin-5-yl-acetic acid (**6b**), 1-methylthiourea (**2a**), methylthiozolidinone (**3a**), methyl-4-bromophenyl-thiazoline (**5a**), and methyl-thiazolidinone-acetic acid (**6a**) had a negative effect on the anticancer activity. 

On the other hand, the second compound series containing 2-benzylidenehydrazinyl-thiazole scaffold showed generally lower activity when compared to the first series. Addition of any substitution on the phenyl ring e.g., 4-methyl (**9b**), 3,4-dimethoxy (**9c**), 4-nitro (**9d**) 4-flouro (**9f**), decreased the anticancer activity. Only the unsubstituted phenyl derivative (**9a**) showed the lowest IC_50_ value against all three cell lines (Table 4). On the other hand, replacement of the phenyl substitution with furan decreased the cytotoxic activity when compared to compound 9d. However, a nitro-furan ring offered relatively good cytotoxic activity when compared to the rest of the compounds in the series (Figure 11).

#### 2.6.2. In Vitro Activity against CDK4 Enzyme

Compounds with the highest in-vitro cytotoxicity were tested for their CDK4 enzyme inhibition. The results are presented in Table 5. Compound **2b** showed the highest inhibitory activity against CDK4 enzyme. 

#### 2.6.3. Radioiodination of Compound 2b

Compound **2b** showed the most promising anticancer activity in the previous *in-vitro* assays. Its tagging with a radiotracer (^131^I) was planned to study its in vivo distribution and selective localization at tumor site. Radioiodination of **2b** was achieved via electrophilic substitution reaction utilizing the oxidative effect of chloramine-t (CAT). To achieve optimized reaction conditions and obtain the highest percentage of **^131^I-2b** (highest radiochemical yield, RCY), various factors were studied. These factors are, the amount of oxidizing agent, the amount of **2b**, pH of the reaction and the reaction time. The amount of CAT used showed to influence the % RCY as depicted in Figure 12A. Starting with only 25 μg of CAT the RCY was 25.74 + 3.23% this yield was increased proportionally with amount of CAT to reach 91.84 + 5.82% when 50 μg of CAT was used. A further increase in the CAT amount caused a significant drop in the RCY, which can be attributed to the formation of chlorinated by-products [59]. Similarly, the amount of **2b** had the same effect on the RCY (Figure 12B). The RCY was only 8.01 + 1.33% when 100 μg of **2b** was used and roused to 91.84 + 5.82% at 500 μg of **2b**. The pH of the reaction was found to be 6 without the use of a buffer solution. In this study dil. HCl and 0.01 M NaOH was used to alter the pH of the reaction. The best RCY (91.84 + 5.82%) was achieved at pH 2. The RCY decrease by increasing the pH towards more basic values to reach only 14.40 + 3.11% at pH 7 (Figure 12C). 

Finally, the reaction time was found to be crucial in obtaining the RCY (Figure 12D). The RCY was 60.09 + 5.11 after 15 min and reached its optimum at 45 min. At longer reaction time the RCY slightly declined to reach 87.2 + 5.05%. 

#### 2.6.4. Chromatographic Determination of the RCY

The RCY was evaluated using thin layer chromatography (TLC). Two mobile phases were used for achieving good separation between **^131^I-2b** and free ^131^I and to ensure accurate determination of the RCY. Chloroform:methanol was used as the first mobile phase; free ^131^I had a retention factor (R*_f_*) of 0.2 and **^131^I-2b** had R*_f_* of 0.5. The second mobile phase was chloroform: ethanol (9:1). The R*_f_* for ^131^I and **^131^I-2b** were 0.3 and 0.5, respectively. A blank reaction was used to establish the R*_f_* of free ^131^I.

#### 2.6.5. Purification of ^131^I-2b by Column Chromatography

The separation of **^131^I-2b** to its precursor **2b** and from other radioactive species was achieved via column chromatography. Silica column was used for the purification process using a mixture of chloroform: methanol (4.8: 0.2) as a mobile phase. The eluate was monitored by TLC. **^131^I-2b** was eluted first (R_t_ = 1 min) followed by ^131^I at 3.5 min (Figure 13). Cold unreacted **2b** appeared at R_t_ 1.75 min as detected by TLC.

#### 2.6.6. In Vitro Stability of ^131^I-2b

To ensure the stability of the radioiodinated compound, column purified **^131^I-2b** was kept in a dark vial to avoid the effect of light. A sample was withdrawn at various time points (2, 4, 6 and 8 h). **^131^I-2b** showed good stability up to 4 h where the change in RCY was only −0.87%. After 12 h and 24 h the change in RCY was −11.82% and −26.21% (Figure 14).

#### 2.6.7. Biodistribution Study

Pure **^131^I-2b** was injected in both normal and solid tumor bearing mice to establish the normal route of **^131^I-2b** distribution and to monitor its localization at tumor site. **^131^I-2b** showed a comparable distribution pattern in both normal and tumor bearing mice. As can be observed the **^131^I-2b** showed rapid clearance from the blood where the initial uptake was 2.71 ± 0.93% injected dose/g (% ID/g) 15 min post injection (p.i) which gradually declined by time to reach a value of 0.4 ± 0.018% ID/g, 240 min p.i. 

The liver and kidney uptake clearly established the metabolic and excretory pathway of **^131^I-2b**. The liver uptake started at 5.20 ± 0.83 at 15 min p.i and started to increase gradually to reach its maximum at 120 min p.i (14.29 ± 2.85%ID/g) followed by a decline in the uptake afterwards as the drug was cleared from the body. Likewise, the kidneys started with low uptake (0.86 ± 0.21%ID/g at 15 min p.i) and started to increase gradually with time to reach a highest of 7.46 ± 1.96%ID/g at 120 min p.i (Figure 15).

The biodistribution of **^131^I-2b** is demonstrated in Figure 16. Thyroid uptake is considered as an indicator for the in vivo stability of radioiodinated compounds. Low thyroid uptake indicates that the compound is resistant to in vivo de-iodination^59^. A low uptake of radioactivity was observed at 15 min p.i. (0.67 ± 0.06% ID/g) which increased to reach a peak of 8.27 ± 1.09% ID/g at 240 min p.i. which indicated gradual in vivo de-iodination of **^131^I-2b_._**

In solid tumor bearing mice, the values of tumor uptake were promising starting with uptake of 4.5 ± 0.54% ID/g at 15 min p.i. which continued to increase to reach a height of 5.97 ± 0.82% ID/g at 60 min p.i. At longer time points, the uptake decreased gradually to reach 0.47 ± 0.15% ID/g. On the other hand, normal muscle showed negligible uptake at all time points (Figure 17).

Tumor: normal tissue ratio is used as a parameter to evaluate preferential uptake and tumor targeting of a radiolabeled agent [60]. The calculated values of ratio of (tumor:blood) (T/B) and ratio of (Tumor: Normal muscle) (T/NM) were; 1.66 and 2.75 at 15 and 30 min p.i, respectively Table 6. Compared with blood, the radioactivity uptake was1.66 and 2.75 at 15 and 30 min p.i, respectively. This ratio reached its peak (3.75) at 60 min p.i. and declined to 0.76 at 120 min p.i. As for T/NM ratio, **^131^I-2b** showed high preferential uptake when compared to normal muscle. The lowest T/NM value (7.12) achieved 120 min p.i. and highest value of 94.04 at 60 min p.i. Comparing T/NM for **^131^I-2b** with other tumor targeting agents, such as ^177^Lu-DTPA-DG (5.07 at 60 min p.i) [61], ^125^I-antiTLR5 mAb (6.481 at 48 h) [62], and ^99m^Tc-dioxime (5.14 at 30 min) [63], indicating its high targeting ability as well as the fact that **^131^I-2b** can be considered as a good radiotheranostic agent for solid tumors.

## 3. In Silico Evaluation of ADMET

Different predictive qualitative ADMET models were applied to study the effect of different physicochemical properties of the synthesized compounds on their expected absorption, metabolism, and toxicity using admetSAR server (Table 6). Except for compound **9i**, all the tested compounds showed the ability to cross the blood brain barrier (BBB). In addition, all the compounds had good predicted human intestinal absorption (HIA). Compounds **3a**, **5a**, and **9i** were predicted to be substrate for CYP450 3A4. On the other hand, all the compounds were predicted to be inhibitors to CYP450 1A2. Compounds **2b**, **9a**, and **9i** showed predicted inhibitory activity for CYP450 3A4. The four compounds showed non-toxic profile as predicted by the studied parameters.

## 4. Materials and Methods

### 4.1. Chemical Synthesis

All melting points are uncorrected and were taken in open capillary tubes using a Stuart melting point apparatus. Elemental microanalyses were carried out at the Regional Center for Mycology and Biotechnology, Al-Azhar University, Cairo, Egypt. Mass spectra (MS) were performed at 70 e.v by GCMS-QP1000 EX spectrometer using the electron ionization technique (EI) at the Regional Center for Mycology and Biotechnology, Al-Azhar University, Cairo, Egypt. Infrared spectra (IR) were recorded (KBr discs) on a Shimadzu FT-IR 8201 PC spectrophotometer, faculty of Science, Cairo University (Cairo, Egypt). NMR spectra were recorded on a mercury spectrometer at 300 MHz or Bruker NMR spectrometer at 400 MHz. ^1^H NMR spectra were run at 300 or 400 MHz, while ^13^C spectra were run at 75 MHz at the Faculty of Pharmacy, Ain shams University, Cairo, Egypt. Chemical shifts were expressed in δ (ppm) downfield from TMS as an internal standard. Follow-up of the reactions and checking the purity of the compounds were made by TLC on silica gel-precoated aluminum sheets (Type 60, F 254, Merck, Darmstadt, Germany) using hexane/ethylacetate (3:1, *v*/*v*) and the spots were detected by exposure to a UV lamp at 254 nanometer for few seconds and by iodine vapor. The chemical names given for the prepared compounds are according to the IUPAC system. 

Synthesis of 2-bromo-1-(1H-benzo[d]imidazol-2-yl)-1-ethanone (3) was achieved according to the reported method [47,50,51,52,53]. 

General procedure for synthesis of 2-(1-(6-bromo-2-oxo-2H-chromen-3-yl)ethylidene)-*N*-methyl/phenylhydrazine-1-carbothioamide **2a,b**

To a hot solution of the 3-acetyl-6-bromo-2H-chromen-2-one (1.8 mmol) in absolute ethanol (10 mL), 4-methyl/phenyl-thiosemicarbazide (1.8 mmol) and five drops of concentrated HCl were added. The mixture was refluxed for 14–17 h. The reaction mixture was allowed to cool, the formed precipitate was filtered, dried and recrystallized from absolute ethanol to afford the title compounds **2a,b**.

General procedure for synthesis of 2-((1-(6-bromo-2-oxo-2H-chromen-3-yl)ethylidene)hydrazino)-3-methyl/phenyl-thiazolidin-4-one **3a,b**

To a hot solution of the **2a,b** (1.4 mmol) in absolute ethanol (20 mL), ethyl bromoacetate (0.2 g, 1.5 mmol) and anhydrous sodium acetate (0.2 g, 2.8 mmol) were added. The reaction mixture was refluxed for 7–12 h. The formed precipitate was filtered, washed with water, dried, and recrystallized from the appropriate solvent to afford the corresponding title compound **3a,b**.

General procedure for synthesis of 6-Bromo-3-(1-((3-methyl/phenyl-4-phenylthiazol-2(3H)ylidene)hydrazineylidene) ethyl)-2H-chromen-2-one **4a,b**

To a hot solution of the 6-bromocoumarin compound **2a,b** (1.4 mmol) in absolute ethanol (20 mL), phenacylbromide (0.3 g, 1.4 mmol) and anhydrous sodium acetate (0.2 g, 2.8 mmol) were added. The reaction mixture was refluxed for 9 h. The formed precipitate was filtered then washed with water, filtered, dried, and recrystallized from the proper solvent to afford the corresponding the title compounds **4a,b**.

General procedure for synthesis of bromo-3-(1-((4-(4-bromophenyl)-3-methyl/phenylthiazol-2(3H)-ylidene) hydrazono)ethyl)-2H-chromen-2-one **5a,b**

To a hot solution of compound **2a,b** (1.2 mmol) in absolute ethanol (20 mL), p-bromophenacylbromide (0.3 g, 1.2 mmol) and anhydrous sodium acetate (0.2 g, 2.4 mmol) were added. The reaction mixture was refluxed for 19 h. After reaction completion, the formed precipitate was filtered while hot then washed with water, filtered, dried, and crystalized from the proper solvent to accomplish the corresponding derivatives **5a,b**.

General procedure for synthesis of 2-(2-((1-(6-bromo-2-oxo-2H-chromen-3-yl)ethylidene)hydrazono)-3-methyl/phenyl-4-oxothiazolidin-5-yl)acetic acid **6a,b**

To a hot solution of compound **2a,b** (1.2 mmol) in toluene (20 mL), maleic anhydride (1.2 mmol) was added. The reaction mixture was refluxed for 16 h till reaction completion, then left to cool. The formed precipitate was filtered, dried, and recrystallized from the proper solvent to afford the corresponding title compounds **6a,b**.

General procedure for synthesis of 1-(substituted) thiosemicarbazides (Schiff bases) **7a**–**i**

The synthesis of thiosemicarbazones was carried out according to the reported procedures 57–63 and the brief procedure is provided in (Appendix A).

General procedure for the synthesis of 3-(2-(2-(substituted)hydrazinyl)thiazol-4-yl)-6-bromo-2H-chromen-2-one **9a**–**i**

6-Bromo-3-(2-bromoacetyl)-2H-chromen-2-one (0.69 g, 0.002 mol) and the appropriate thiosemicarbazone (0.002 mol) were dissolved in ethanol (10 mL). The reaction mixture was refluxed for 5–7 h, monitored by TLC. The formed precipitate was filtered while hot, dried and recrystallized from the proper solvent to afford the corresponding title compounds **9a**–**i**, respectively.

### 4.2. QSAR Study

All structures were built and saved in a database (mdb format). The *pIC*_50_ were computed. The energy of built compounds was minimized. MOE 2016.08 molecular descriptors were calculated for all compounds. The previously reported 16 synthesized compounds [47] were used as a training set and our newly synthesized compounds were considered as a test set. A QSAR model was created. All redundant descriptors with low variance among the different compounds were removed. Multiple linear regression was used as the machine learning algorithm for linear model generation which removes co-linear descriptors by default. 

### 4.3. Molecular Docking

The crystal structure of CDK4 in complex with flavopiridol that has been resulted from superimposition of (CDK2 + Flavopiridol) (pdb code = 2W9Z) and CDK4 (pdb code = 6GUB). The resulted (CDK4 + Flavopiridol) model was used for molecular docking and the protein was prepared prior to the docking process by MOE 2016.08 in which minimization were done. Hydrogen atoms were added, and the docking site was identified and selected. All compounds were built and saved as (moe). Rigid receptor was used as a docking protocol. Both receptor and solvent were kept as a “receptor”. Triangle matcher was used as a placement method. Two rescoring were computed, rescoring 1 was selected as London dG. Rescoring 2 was selected as affinity. Force field was used as a refinement. All coordinates were derived from pdb and all interactions were observed with the conserved residues. The docking protocol used the triangle method as a placement method with timeout of 300 s, and number of return poses as 1000. London dG was used as a rescoring method. Force field was used as a refinement method by applying MMFF94x.

### 4.4. Molecular Dynamic Simulations

All molecular dynamic simulations were conducted by MOE 2016.08. The best conformations from each docking process were kept inside the active site. The quality of the temperature-related factors, protein geometries, and electron density were tested. All hydrogens were added, and energy minimization was computed. The solvent molecules that were in the system were deleted before solvation and salt atoms were added to ensure complete neutralization of the biomolecular system. The solvent atoms were added to surround the biomolecular system (protein-ligand complex) in a spherical shape. The protein-ligand complex was surrounded by a sphere shape of solvent (water) and NaCl was used as a salt to neutralize charged system. Amber 10:EHT was selected as a force field in the potential setup step. All Van der Waals forces, electrostatics, and restraints were enabled. The heat was adjusted in order to increase the temperature of the system from 0 to 300 K which was followed by equilibration and production for 300 ps. The simulation was conducted over a 10 ns period of time. 

### 4.5. Cytotoxic Evaluation

Mammalian cell lines: MCF-7 cells (human breast cancer) cell line, A-549 (human Lung Carcinoma) and CHO-K1 cells (hamster ovary cancer) cell line were obtained from the American Type Culture Collection (ATCC, Rockville, MD). Fetal bovine serum, RPMI-1640, HEPES buffer solution, L-glutamine, gentamycin and 0.25% Trypsin-EDTA were purchased from Lonza (Belgium). All reagents and chemicals were of pharmaceutical pure grade. Non-carrier added Na^131^I solution (pH = 7) was purchased from the Research and Production Facility, The Egyptian Atomic Energy Authority. Chemicals were purchased from Sigma Aldrich and Lonza. Monitoring of chemical reactions was done by using analytical TLC with Merck 60 F-254 silica-gel plates (Merck, Germany), and visualization was done with ultraviolet light. 

Cytotoxicity study was performed at the Regional Centre for Mycology and Biotechnology (RCMB), Al-Azhar University. Mammalian cell lines: human breast cancer cell line (MCF-7 cells), human colon cancer cell line (HCT-116 cells), and human lung cancer cell line (A-549 cells) were obtained from VACSERA Tissue Culture Unit, Cairo, Egypt. The cells were propagated in Dulbecco’s modified Eagle’s medium supplemented with 10% heat-inactivated fetal bovine serum, 1% L-glutamine, HEPES buffer, and 50 μg/mL gentamycin. All cells were maintained at 37 °C in a humidified atmosphere with 5% CO_2_ and were subcultured two times a week. For the cytotoxicity assay, the cells were seeded in 96- well plate at a cell concentration of 1 × 104 cells per well in 100 μL of growth medium. Fresh medium containing different concentrations of the test sample was added after 24 h of seeding. Serial two-fold dilutions of the tested chemical compound were added to confluent cell monolayers dispensed into 96-well, flat-bottomed microtiter plates (Falcon, NJ, USA) using a multichannel pipette. The microtiter plates were incubated at 37 °C in a humidified incubator with 5% CO_2_ for a period of 48 h. Three wells were used for each concentration of the test sample. Control cells were incubated without test sample and with or without DMSO. The little percentage of DMSO present in the wells (maximal 0.1%) was found not to affect the experiment. After incubation of the cells for at 37 °C, various concentrations of sample were added, and the incubation was continued for 24 h, and viable cells yield was determined by a colorimetric method. In brief, after the end of the incubation period, media were aspirated, and the crystal violet solution (1%) was added to each well for at least 30 min. The stain was removed, and the plates were rinsed using tap water until all excess stain is removed. Glacial acetic acid (30%) was then added to all wells and mixed thoroughly, and then the absorbance of the plates was measured after gently shaken on a microplate reader, using a test wavelength of 490 nm. All results were corrected for background absorbance detected in wells without added stain. Treated samples were compared with the cell control in the absence of the tested compound. All experiments were carried out in triplicate. The cell cytotoxic effect of each tested compound was calculated. The optical density was measured with the microplate reader to determine the number of viable cells, and the percentage of viability was calculated as (1 − (ODt/ODc)) × 100% where ODt is the mean optical density of wells treated with the tested sample and ODc is the mean optical density of untreated cells. The relation between surviving cells and drug concentration is plotted to get the survival curve of each tumor cell line after treatment with the specified compound. The 50% inhibitory concentration (IC_50_), the concentration required to cause toxic effects in 50% of intact cells, was estimated from graphic plots of the dose response curve for each conc [64,65].

### 4.6. CDK4 Enzyme Inhibition Assay

Reaction Biology Corp. Kinase HotSpotSM service (http://www.reactionbiology.com, accessed on 1 April 2019) was used for screening of tested compounds. CDK4 Kinase inhibitory activities were assessed by the HotSpot assay platform, which contained specific kinase/substrate pairs along with required cofactors. Base reaction buffer: 20 mM Hepes (pH 7.5), 10 mM MgCl2, 1 mM EGTA, 0.02% Brij35, 0.02 mg/mL BSA, 0.1 mM Na3VO4, 2 mM DTT, 1% DMSO. Testing compounds were dissolved in 100% DMSO to specific concentration. The serial dilution was conducted in DMSO. The reaction mixture containing the examined compound and 33P-ATP was incubated at room temperature for 2 h and radioactivity was detected by the filter-binding method. Kinase activity data were expressed as the percent remaining kinase activity in test samples compared to vehicle (dimethyl sulfoxide) reactions.

For CDK4 enzyme inhibition, CDK4 ELISA kit, Cloud-Clone Corp, US was used. This kit is used for the in vitro quantitative measurement of CDK4 concentrations in tissue homogenates, cell lysates and other biological fluids with high sensitivity and specificity. The cancer cells were treated with different amounts of each compound according to the aforementioned protocol. In addition, the cells were lysed and then treated as instructed by the manufacturer’s protocol. The absorbance of each well was detected at 450 nm using an ELISA plate reader. The results are reported as half maximal inhibitory concentration (IC_50_). 

### 4.7. Radioiodination of **2b**

Stock solutions of **2b**, Chloramine-T (CAT), and sodium thiosulfate were prepared for use in the radioiodination reaction as follows. Amount of 5 mg of **2b** was dissolved in 5 mL of DMF. For CAT stock solution, exactly 1 mg of CAT was dissolved in 1 mL absolute ethanol. For sodium thiosulfate stock solution; amount of 15 mg sodium thiosulfate was dissolved in 1 mL of double distilled water. Radioiodination reaction was prepared as follows; 500 μL (1.20 μM) of **2b** stock solution was placed in a dark screw cap vial followed by the addition of 10 μL of Na^131^I solution (equivalent to 150 MBq) then exactly 50 μL of CAT solution was added (0.22 μM) the reaction pH was adjusted to 2 using 80 μL of dil HCl. The reaction mixture was vortexed and allowed to stand for 45 min at room temperature. Afterwards the reaction was stopped using 5 μL of sodium thiosulfate solution.

### 4.8. Determination of the Radiochemical Yield

The radiochemical yield was determined using paper chromatography using TLC (1 cm × 13 cm strip). Exactly 20 μL (4.6 MBq) of the reaction mixture was placed 1 cm away from the TLC plate base and was developed using chloroform: methanol (4.8: 0.2) and chloroform:ethanol (9: 1). After complete development, the TLC plate was air dried and cut into 1 cm strips. Each strip was counted in a well type γ-counter. A blank reaction was analyzed similarly to establish the retention factor (Rf) of ^131^I.

### 4.9. Purification of **^131^I-2b**

For the purpose of purification, a silica column was used (1 cm × 10 cm). Amount of 100 μL (23.25 MBq) was injected to the column and eluted using chloroform: methanol (4.8: 0.2). The flow rate was 0.5 mL/min and eluted fractions were collected manually and counted individually using well type γ-counter. All the eluted fractions were also analyzed individually by TLC.

### 4.10. In-Vitro Stability Study

Column purified **^131^I-2b** was kept in a dark screw vial in dark chamber for 24 h. A sample of the solution was withdrawn and the radiochemical yield was re-evaluated by TLC at different time points (2, 4, 6, 12, 24 h).

### 4.11. Biodistribution Study

Ehrlich ascites carcinoma (EAC) was used in this study. The parent line was supplied from the ascitic fluid of Swiss Albino mice purchased from the Egyptian National Cancer Institute, Cairo University. Adult male Swiss albino mice weighing 22–25 g purchased from the breeding unit of the Egyptian Organization for Biological Products and Vaccines, Cairo, Egypt, were used in this study. Animals were on pellet diet and tap water. Animal treatment was in accordance with the National Institute of Health Guide for Animal, as approved by the Institutional Animal Care and Use Committee. The EAC was withdrawn from the ascitic fluid of a mice bearing EAC (10 days old) under aseptic conditions and diluted using physiological sterile saline solution. The diluted cell line was mixed for 5 min by vortex shaker. To generate Ehrlich solid tumor, 200 μL of the diluted ascitic fluid was injected intramuscularly in the right lower limb of male mice (22–25 g). Palpable solid mass was observed 10 days after inoculation.

For commencing the biodistribution study the animals were divided into two groups the first included normal mice (no tumor inoculation) and the second group included tumor inoculated mice. Five individuals were used for each time point. Exactly 100 μL (23.25 MBq) of purified **^131^I-2b** was injected in the tail vein of each mice. Mice were anesthetized with a mixture of xylazine (10 mg/kg) and ketamine (80 mg/kg) then weighed and dissected. Blood was collected via cardiac puncture, and all organs were separated and washed with physiological saline. For the tumor muscle, sample of the tumor was isolated completely from the leg and washed with normal saline then weighed. Another sample of the contralateral normal muscle was used as control to measure the tumor uptake of compound **2**. All the collected biological samples were placed in previously weighed plastic containers, and the activity was counted in automatic scintillation gamma counter. A standard dose containing the same injected amount was counted simultaneously and accounted as 100% activity [66] The uptake was expressed as percentage injected dose/gram tissue (%ID/g). Blood, bone, and muscles were assumed to be 7%, 10%, and 40% of the total body weight, respectively [67]. Data are reported as the mean ± SE. Data were analyzed using compare means function in SPSS software (Version 20.0).

### 4.12. In Silico Prediction of ADMET

Different absorption, distribution, metabolism, excretion and toxicity parameters were evaluated using the qualitative models offered by the free webserver admetSAR (http://lmmd.ecust.edu.cn/admetsar2) {Yang, 2018 #245}. The compounds were converted to smiles format and uploaded to obtain the desired prediction.

## 5. Conclusions

This study showed the discovery of novel coumarin-based lead compound with promising anticancer and selective radiotheranostic activity. Compound **2b** had potential cytotoxic activity with IC_50_ = 0.0136, 0.015, 0.054 µM against MCF-7, A-549, and CHO-K1 cell lines, respectively. Its binding stability within CDK4 was confirmed by both docking (ΔG = −15.34 kcal/mol) and dynamic simulations in which it achieved equilibrium at RMSD of 0.8 Å after 3.5 ns. This was confirmed by the CDK4 enzyme assay where **2b** showed inhibitory activity of CDK4 = 0.036 µM. In addition, the radio-iodinated form of compound **^131^I-2b** showed good in vitro stability up to 4 h, biodistribution pattern, tumor uptake reached a height of 5.97 ± 0.82%ID/g at 60 min p.i and promising results in terms of selective targeting and localization in solid tumors. A finding that will encourage to use compound **2b** as a chemotherapeutic agent, a radiotherapeutic agent, and a radio-imaging agent for solid tumors.

## Data Availability

Not applicable.

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
