# Peer review of "Discovery of New Coumarin-Based Lead with Potential Anticancer, CDK4 Inhibition and Selective Radiotheranostic Effect: Synthesis, 2D & 3D QSAR, Molecular Dynamics, In Vitro Cytotoxicity, Radioiodination, and Biodistribution Studies"

_molecules, 2021, doi:10.3390/molecules26082273_

Round 1

Reviewer 1 Report

Authors present on a first sight a comprehensive study focused on the development of new inhibitors of cyclin dependent kinase 4 that could be potentially used as cancerostatic treatment. Authors tried to employ a large variety of methods ranging from synthesis, through in silico characterisation to in vitro testing. This could be potentially an interesting and usefuls study, however, most of the approaches seems to be performed and described in very low detail with a very low feeling for science and actual sense for analytical thinking.

I will refer mostly to the in silico part, which is my major area of interest.

In current format the paper is unacceptable. Molecular dockig as well as chemometric analysis is given with too little detail and the reader can have feeling that "some kind of in silico analysis has been done and it is OK". QSAR is very old and criticised approach, exactly because of the uncritical use like here. Authors found a chance correlation of affinity data with one descriptors out of many (this is always the case - one descriptor will always show chance correlation). They do not explain properly why and how the QSAR model is supposed to work. Nevertheless, they use it to judge the compounds.

Molecualr Docking is described by a few sentences and is performed with little respect for detail so that others could repeat/reproduce the results. Docking is an extremely sensitive approach and thus if used in a discussion, it must be done with best possible accuracy and treatment of all possible factors influencing the result. Authors claim they did docking to a homology structure, but they do not give almost any detail on how this was done... Again, this is a very tricky procedure and can in many aspects mislead if not done properly. As far as one can see from the figures, some rings of docked poses are distorted... Authors also do not discuss why they chose exactly CDK4 - as far as I can see the lead compound flavopiridol is cocrystalized with further targets and it could be interesting to see thedocking modes of cnewly syntesized compounds to those targets (BRD4, glycogen phosphorylase) too.

Molecualr dynamics simulations are also very poorly described - I was surprized to read that authors cooled the MD simulation to 0 kelvin - normally this is done at 300 K...

I would suggest authors to read a few high impact publications in their field and see in how much detail methods the approaches should be described.

And then the same applies for the discussion. Results need to be put to critical, but logical relationship and context, citing and stating exact scientific statements.

In the abstract the correlation coefficient is given without upper index and misses decimal point... This shows that authors paid too little attanetion before sending out the manuscript... Affinity is given in a decimal number without units... "and its affinity 40.05." ???

Reviewer 2 Report

Manuscript ID: molecules-1163660

Title: Discovery of new coumarin-based lead with potential anticancer, CDK4

inhibition and selective radiotheranostic effect: Synthesis, 2D & 3D

QSAR, Molecular dynamics, in vitro cytotoxicity, radioiodination and

biodistribution studies.

Authors: Mona O Sarhan, Somia S Abd El-Karim, Manal M Anwar, Raghda H Gouda,

Wafaa A Zaghary *, Mohammed A Khedr *

Submitted to section: Medicinal Chemistry,

This publication is proposed to Medicinal Chemistry and I think the widely described synthesis part with the description of spectroscopic data should be instanced in supporting information together with a description of the other recipe.

The introduction is too long. The whole publication is hard to read, there is too much information that is unnecessary, which is hard to make sense of the whole concept. Maybe the publication should be divided into two parts.

Abstract:

Line 17:  6-bromo-chomarin-ethylidene I think should be   6-bromo-coumarin-ethylidene

Line 18 analogously

Line 19: 1H- NMR, 13C-NMR should be 1H NMR, 13C NMR,

In my opinion the abstract is too long and includes too much information, there should be made a selection  for a main  sentence

Line 70: cells cell cycle?

Line 102: Using various Structural modifications, Banyu 102 Pharmaceutical Co., Ltd [40]. discovered the 2-aminothiazole compounds?

Line 149: 1H NMR, 13C NMR?

Line 152: Scheme 1. Synthesis of new 6-bromo-chomarin-ethylidene hydrazonyl-thiazolyl derivatives the name is not correct

Line 154: Scheme 2. Synthesis of new 6-bromo-chomarin – thiazolyl-based derivatives - the name is not correct

Line 156: thiazol-hydrazono-coumarin should be thiazolyl-hydrazono-coumarin

Line 159: Quantitative structure activity relationship (QSAR) study of this series was also performed and reported and revealed that the hidden features that contributed to the biological activity were; the molecular partial charge distribution and hydrophilicity. – this is hard to read because of the repeated words

Line 163: thiazol-hydrazono coumarin should be thiazolyl-hydrazono-coumarin

Line 183:  Figure 6. Best docking poses of compounds: A) 2b pose I, B) 2b pose II, C)3b D) 5a. I think that is C and D is something omitted.

Line 203: Table 3. Cytotoxic effect of the scheme I is written in the whole text like this but in  under scheme 1 like this.  

131I-2b is written this way but should be written like this 131I-2b

Line 268: appeared δ 29.91 ppm should be appeared at δ 29.91 ppm

Line 298: arylidenehydrazine-1-carbothioamides 7a-I 57-63 what does it mean?

Line 303: 33418 - 3503 cm−1 and 1675-1728 cm−1 should be 33418 - 3503 cm−1 and 1675-1728 cm−1

Line 329: Table 7. Prediction of the pIC50 values for the synthesized compounds 4. Materials and Methods. ?

Line 368: 2-benzylidenehydrazineyl-thiazole scaffold should be 2-benzylidenehydrazinyl-thiazole scaffold

Line 432: vivo de-iodination59 what does it mean? If it is literature it should be in a bracket.

Line 441: of a radiolabeled agent 60?

Line 443: was1.66?

Line 492: CH3 should be CH3 in the whole experimental part

Line 497: For C13H12BrN3O2S should be  For C13H12BrN3O2S in the whole experimental part

  1. Zoltewicz JA, Deady LW. Quaternization of heteroaromatic compounds: quantitative aspects, Adv. Heterocycl. Chem. 1978; 22: 1023 71–121. 1024

58- Molecular Operating Environment (MOE), 2016.08; Chemical Computing Group Inc., 1010 Sherbrooke St. West, Suite #910, Mon-1025 treal, QC, Canada, H3A 2R7, 2016  why is the literature written in a different way?

That is only a part of the mistakes in this manuscript. The whole text needs careful attention and editorial correction.  

Reviewer 3 Report

The manuscript might be accepted after some revisions.

The main concern regards the editing and organization of the manuscript. In fact, I strongly recommend to unify Results and Discussion in one more organized paragraph. In my opinion, all subparagraphs of the section Results, several of which have only the title and a figure, should be removed and included in only one paragraph subdivided into subparagraphs in which the discussion of the obtained results would be improved. The proximity between text and figures to which the text refers would also be necessary to facilitate the reading of the manuscript. In addition, I recommend to indicate the tested compounds with their numbering instead of the scheme in which their synthesis is described (see title of paragraph 3.6.1 and 3.6.2).

Another question is if compound 131I-2b has been characterized by means of spectroscopy; its structure should also be included in a figure to indicate the position of 131I.

Finally, it should be recommended to carefully revised the text in order to improve English language and to correct several oversights.
